# Molecular Mechanisms Underlying TNFα-Induced Mitochondrial Biogenesis in Human Airway Smooth Muscle

**DOI:** 10.3390/ijms24065788

**Published:** 2023-03-17

**Authors:** Debanjali Dasgupta, Sanjana Mahadev Bhat, Alexis L. Price, Philippe Delmotte, Gary C. Sieck

**Affiliations:** Department of Physiology & Biomedical Engineering, Mayo Clinic, Rochester, MN 55905, USA

**Keywords:** TNFα, airway inflammation, mitochondrial biogenesis, CREB, ATF1, PGC1α, mtDNA, NRFs

## Abstract

Proinflammatory cytokines such as TNFα mediate airway inflammation. Previously, we showed that TNFα increases mitochondrial biogenesis in human ASM (hASM) cells, which is associated with increased PGC1α expression. We hypothesized that TNFα induces CREB and ATF1 phosphorylation (pCREB^S133^ and pATF1^S63^), which transcriptionally co-activate PGC1α expression. Primary hASM cells were dissociated from bronchiolar tissue obtained from patients undergoing lung resection, cultured (one–three passages), and then differentiated by serum deprivation (48 h). hASM cells from the same patient were divided into two groups: TNFα (20 ng/mL) treated for 6 h and untreated controls. Mitochondria were labeled using MitoTracker green and imaged using 3D confocal microscopy to determine mitochondrial volume density. Mitochondrial biogenesis was assessed based on relative mitochondrial DNA (mtDNA) copy number determined by quantitative real-time PCR (qPCR). Gene and/or protein expression of pCREB^S133^, pATF1^S63^, PCG1α, and downstream signaling molecules (NRFs, TFAM) that regulate transcription and replication of the mitochondrial genome, were determined by qPCR and/or Western blot. TNFα increased mitochondrial volume density and mitochondrial biogenesis in hASM cells, which was associated with an increase in pCREB^S133^, pATF1^S63^ and PCG1α expression, with downstream transcriptional activation of *NRF1, NRF2*, and *TFAM.* We conclude that TNFα increases mitochondrial volume density in hASM cells via a pCREB^S133^/pATF1^S63^/PCG1α-mediated pathway.

## 1. Introduction

Airway inflammation is one of the major contributors to the pathophysiology of airway diseases including asthma [1,2], chronic obstructive pulmonary disease (COPD) [3,4], chronic bronchitis [5], and COVID-19 [6,7,8]. Previous studies, including those from our laboratory, have shown that acute exposure to proinflammatory cytokines such as TNFα results in airway smooth muscle (ASM) hyperreactivity in response to agonist stimulation, leading to energetic stress [9,10]. In addition, TNFα induces the proliferation of ASM cells, which also increases energy demand [11].

In ASM cells, as in other eukaryotic cells, ATP demand is met by mitochondrial respiration and oxidative phosphorylation [11]. Previous studies from our laboratory have reported that in human ASM (hASM) cells, mitochondrial volume density and mitochondrial biogenesis, denoted by relative mitochondrial DNA (mtDNA) copy number are increased in response to TNFα exposure for 24 h, with a corresponding increase in the expression of peroxisome proliferator-activated receptor gamma coactivator 1-alpha (PGC-1α) [11]. Other studies have shown that PGC1α acts as a transcriptional coactivator that binds to the regulatory region of nuclear respiratory factors (NRFs) within the cell’s nucleus [12,13,14,15,16,17]. Downstream, NRFs activate transcription of transcription factor A mitochondrial (TFAM) [16,17], which promotes the replication and transcription of the mitochondrial genome [18,19].

The cAMP-response element binding protein (CREB) is a nuclear bZIP (basic leucine zipper) family of proteins that acts as a transcription factor upon phosphorylation at its specific serine residue 133 (Ser133) [20,21,22]. Phosphorylated CREB (pCREB^S133^) specifically binds to the cAMP response element (CRE) present in the promoter region of its target genes and initiates transcription [20,21,22]. In human hepatocytes [23] and human skeletal muscle [24], it has been shown that pCREB^S133^ binds to a conserved CREB binding site within the PGC1α promoter region, thereby increasing the transcription of the PGC1α gene. Other studies have shown that TNFα induces a transient increase in pCREB^S133^ in various tissue types including the optic nerve [25] and endothelial cells [26,27]. To date, no study in hASM cells has explored the role of pCREB^S133^ in the transcriptional activation of PGC1α. Activating transcription factor 1 (ATF1) has a similar sequence to CREB with a homologous phosphorylation domain [28,29,30]. Phosphorylation of pCREB^S133^ is often associated with phosphorylation of ATF1 at serine 63 (pATF1^S63^), and together, pCREB^S133^ and pATF1^S63^ have been reported to function as transcriptional co-activators for downstream gene targets [28,29]. However, whether pATF1^S63^ transcriptionally activates PGC1α has not been explored. As shown in Figure 1, we hypothesize that TNFα induces pCREB^S133^/pATF1^S63^ phosphorylation in hASM cells transcriptionally co-activating PGC1α, which in turn transcriptionally activates expression of downstream NRFs and TFAM, ultimately leading to mtDNA replication and mitochondrial biogenesis.

## 2. Results

### 2.1. Dissociated Bronchiolar Cells Exhibited hASM Phenotype

Before experimentation, dissociated cells were phenotyped based on the expression of α-SMA by immunocytochemical analysis. In accordance with our previously published studies, hASM cells expressing α-SMA uniformly were larger and displayed a fusiform shape. In each of the patient samples used in this study, hASM cells expressing α-SMA were significantly more prevalent among dissociated cells constituting ~95% of the total dissociated cell population (Figure 2A). This finding was further validated by Western blots for α-SMA in cell homogenates (Figure 2B, Appendix A). However, bronchiolar samples from three patients among the nine initial patient samples displayed no detectable α-SMA protein expression relative to total protein and were thus excluded from subsequent analyses. Therefore, confirmed hASM cell samples from six patients were used in all but two experiments. In both mitochondrial imaging and ChIP analyses, samples from only five patients were used due to an inadequate number of hASM cells within a priori criteria of three passages.

### 2.2. TNFα Increased Mitochondrial Volume Density in hASM Cells

In the imaging studies, only larger fusiform shaped hASM cells were included in the analysis of mitochondrial volume density. Mitochondrial volume density was measured in hASM cells treated with TNFα for 6 h (Figure 3A,B) and compared to untreated time-matched controls. After 6 h exposure to TNFα, mitochondrial volume density was significantly greater in hASM cells compared to untreated control hASM cells (* *p* < 0.05) (Figure 3C–E). There was no effect of TNFα exposure on total hASM cell volume [11]. In Figure 3C, data from individual cells (six cells from five patients, *n* = 30 cells) showed that 6 h of TNFα treatment increased mitochondrial volume density compared to untreated control cells. In each patient, mitochondrial volume density in TNFα-treated hASM cells was significantly greater compared to untreated control cells (Figure 3D). Figure 3E summarizes the results of TNFα treatment compared to untreated control hASM cells across all five patients indicating an ~3.4-fold increase in mitochondrial volume density in TNFα-treated cells (* *p* < 0.05).

### 2.3. TNFα Increases mtDNA Copy Number and Mitochondrial Biogenesis in hASM Cells

In the present study, mitochondrial biogenesis was reflected by an increase in mtDNA copy number relative to nuclear DNA. After 6 h of TNFα exposure, there was a significant increase in mtDNA copy number relative to nuclear DNA in hASM cells, indicating an increase in mitochondrial biogenesis (* *p* < 0.05; Figure 4A,B). The expressions of two human mitochondrial genes, *ND1/SLCO2B1* and *ND5/SERPINA1*, were used as a measure for mtDNA copy number. After normalization with nuclear genes *SLCO2B1* and *SERPINA1*, both *ND1* and *ND5* levels were significantly elevated in TNFα-treated hASM cells by ~1.2-fold (* *p* < 0.05 for each gene, *n* = 6 patient samples). In Figure 4A, the expression of both mtDNA genes increased in TNFα-treated cells in all six patients, although in one patient, only a minimal increase for each gene was observed. Figure 4B summarizes results for all six patient samples, showing that TNFα exposure induced an overall ~15% increase in the relative copy number of the two human mtDNA genes *ND1* and *ND5* compared with untreated hASM cells (* *p* < 0.05).

### 2.4. TNFα-Induced pCREB^S133^ and pATF1^S63^ Phosphorylation

In hASM cells, 6 h exposure to TNFα increased pCREB^S133^ phosphorylation by ~1.7-fold (* *p* < 0.05, *n* = 6 patient samples) compared to untreated control hASM cells (Figure 5A–C, Appendix A). Figure 5B compares pCREB^S133^ levels in hASM cells exposed to 6 h TNFα versus untreated control cells from all six patients. Note that TNFα increased pCREB^S133^ levels in all six patient hASM samples, although the effects varied across patient samples. Figure 5C summarizes the results from all six patients showing an overall ~69% increase in pCREB^S133^ phosphorylation in TNFα-treated hASM cells compared to untreated controls (* *p* < 0.05).

The antibody (Table 1) used for detecting pCREB^S133^ cross-reacts and detects the phosphorylated form of activating transcription factor 1 (pATF1^S63^) (Figure 5A). After 6 h TNFα exposure, pATF1^S63^ levels increased in hASM cells compared to untreated control cells from all six patients (Figure 5D; * *p* < 0.05, *n* = 6 patient samples). Figure 5E summarizes the results from all six patients, showing an ~1.7-fold increase in pATF1^S63^ levels in TNFα-treated compared to untreated control hASM cells (* *p* < 0.05, *n* = 6 patient samples). Total CREB levels were not changed after TNFα treatment (Appendix A).

### 2.5. pCREB^S133^ Transcriptionally Activates PGC1α in TNFα-Treated hASM Cells

To investigate the relationship between pCREB^S133^ and mitochondrial biogenesis, bioinformatics analysis was performed to identify the pCREB^S133^ binding site in the regulatory region of *PGC1*α gene, a key regulator of mitochondrial biogenesis. In silico analyses using TF-bind and EPD revealed the presence of a putative pCREB^S133^ binding site in the promoter region of *PGC1*α gene (Figure 6A) at −129 bases upstream to the translational start site. The EPD database also detected a binding site for pATF1^S63^ at the same location. Due to the sequence homology and presence of homologous phosphorylation domain, pATF1^S63^ acts as co-transcriptional activator with pCREB^S133^ for several genes and cell signaling pathways [30,31]. To examine the binding of pCREB^S133^ to this putative binding site, ChIP assay was performed utilizing the hASM cells from five patients treated with TNFα for 6 h compared to untreated time-matched control cells from the same patients. qPCR amplification of the immunoprecipitated chromatin from ChIP revealed that pCREB^S133^ bound to the putative binding site in the *PGC1*α promoter in all five samples (Figure 6B). Interestingly, pCREB^S133^ occupancy at the *PGC1*α promoter was increased by ~2.3 fold in TNFα-treated hASM cells, which was in proportion to the enhanced phosphorylation of pCREB^S133^ (Figure 6B-D, * *p* < 0.05; *n* = 5 patient samples).

### 2.6. TNFα Increases the Expression of PGC1α in hASM Cells

As the major regulator of mitochondrial biogenesis, the protein expression of PGC1α was quantified. In 6 h TNFα-treated hASM cells, PGC1α expression was increased by ~2-fold (* *p* < 0.05; *n* = 6 patient samples) compared with untreated control hASM cells (Figure 7A–C). Figure 7B shows that PGC1α was elevated in hASM cells after TNFα exposure compared to untreated control cells in all six patients. Figure 7C summarizes results from all six patients, showing that 6 h TNFα exposure increased PGC1α expression by ~98% in hASM cells compared to untreated control cells (* *p* < 0.05).

### 2.7. PGC1α Transcriptionally Activates Expression of Downstream Gene Targets

In cells, PGC1α mediates the activation of a downstream signaling cascade involving expression of *NRF1*, *NRF2* and *TFAM* as major regulators of transcription and replication of the mitochondrial genome [18,32,33]. In hASM cells from each patient sample, 6 h TNFα exposure increased mRNA expression of *NRF1, NRF2* and *TFAM* compared to untreated cells from the same patient (Figure 8A,C,E; * *p* < 0.05; *n* = 6 patient samples). Overall, across all six patient samples, there was an ~89% increase in NRF1 mRNA expression after 6 h TNFα treatment (Figure 8B; * *p* < 0.05), an ~71% increase in *NRF2* mRNA expression (Figure 8D; * *p* < 0.05), and an ~87% increase in *TFAM* mRNA expression (Figure 8F; * *p* < 0.05). NRF1 protein levels were also increased after 6 h exposure to TNFα (Appendix A; * *p* < 0.05; *n* = 6 patient samples). Summarizing the data from six patients, NRF1 protein levels increased by ~1.12 fold in TNFα-treated hASM cells (Appendix A; * *p* < 0.05; *n* = 6 patient samples).

## 3. Discussion

The key findings of the present study indicate that TNFα induces an increase in mitochondrial biogenesis (relative mtDNA copy number) and mitochondrial volume density in hASM cells via a pCREB^S133^/pATF1^S63^/PGC1α dependent pathway. We found that 6 h exposure to TNFα-induced pCREB^S133^ and pATF1^S63^ phosphorylation, transcriptional activation of the *PGC1*α gene, and an increase in PGC1α protein expression. Downstream, we found that the TNFα-induced elevation of PGC1α transcriptionally activates gene expression of *NRF1*, *NRF2,* and *TFAM*, the genes responsible for mtDNA replication, transcription, and mitochondrial biogenesis.

### 3.1. TNFα Increases Mitochondrial Volume Density in hASM Cells

The results of the present study confirmed the results of a previous study from our lab, which reported that 24 h TNFα exposure increases mitochondrial volume density in hASM cells [11]. In the present study, hASM cells were treated with TNFα for only 6 h; however, the increase in mitochondrial volume density was comparable to that found after 24 h exposure. As in our previous study, mitochondria were labeled with MitoTracker Green and imaged in 3D using confocal microscopy. Previously, we compared the use of 3D confocal imaging to determine mitochondrial volume density to the use of electron microscopy (EM) used as a gold standard [34,35]. We found that MitoTracker labeling and 3D confocal imaging yielded mitochondrial volume densities in hASM cells that were comparable to EM. However, the confocal imaging technique has the major advantage of mitochondrial volume density measurements in many more cells [34,35]. The 3D confocal imaging technique has the additional advantage over conventional 2D EM techniques that the orientation of mitochondria within the cell does not affect mitochondrial volume estimates [11,36].

### 3.2. TNFα Increases Mitochondrial Biogenesis in hASM Cells

Mitochondrial biogenesis defines the process by which mitochondrial DNA replicates, and if greater than the rate of mitochondrial degradation (via mitophagy), leads to an increase in mitochondrial volume density. Mitochondrial biogenesis is a multistep process involving the synthesis of new mtDNA that translates into mitochondrial protein. In the present study, we measured the mtDNA copy number of two genes normalized to nuclear DNA as a reflection of mitochondrial biogenesis. The results of the present study are consistent with those of a previous study in which we found that 24 h TNFα exposure increases mtDNA copy number relative to nuclear DNA (mitochondrial biogenesis) in hASM cells [11]. The results of the present study extend these previous findings by showing that an increase in mitochondrial biogenesis occurs in hASM cells after only 6 h TNFα exposure, although the increase in mtDNA copy number after 6 h TNFα exposure was less pronounced than that observed after 24 h TNFα exposure [11]. As we previously found, the TNFα-induced increase in mtDNA copy number was consistent with an increase in PGC1α expression and mitochondrial volume density in hASM cells [11].

### 3.3. TNFα Increases pCREB^S133^ and pATF1^S63^ Phosphorylation in hASM Cells

The results of the present study showed that 6 h TNFα exposure increased CREB phosphorylation at Ser133 (pCREB^S133^). pCREB^S133^ is a transcription factor localized to the nucleus and activated in response to a variety of metabolic, trophic, and inflammatory inducers [20,22]. After phosphorylation, pCREB^S133^ binds to the CRE sequence, a highly conserved nucleotide sequence found in the upstream promoter region of its target gene, where it plays a significant role in activating the transcription of target genes. In this transcriptional pathway, pCREB^S133^ associates with other co-activators and adaptor proteins, such as pATF1^S63^ and CREB-binding protein (CBP), and thereby induces the transcription of its target genes [30,31,37]. Previously published studies reported that pro-inflammatory cytokines such as TNFα play a significant role in pCREB^S133^ activation [27,38,39]. Consistent with the results of the present study in hASM cells, it was previously reported that TNFα induced a transient increase in pCREB^S133^ phosphorylation in the optic nerve [25]. In human umbilical vein endothelial cells (HUVEC), TNFα induces pCREB^S133^ via mitogen-activated protein kinase (p38-MAPK)-mediated pathway, with downstream binding to CRE and transactivation of target genes [26]. The phosphorylation of pCREB^S133^ is mediated by several calcium dependent serine––threonine kinases such as cAMP-dependent protein kinase A (PKA), protein kinase C (PKC; including PKCε), and calmodulin kinases (CaMKs; e.g., CaMK-IV) that respond to calcium fluxes from the extra-cellular domain or the intracellular calcium stores [20,27,37,40,41,42]. Previous studies from our lab and other labs demonstrated that TNFα exposure increases cytosolic calcium and maintains agonist-mediated calcium homeostasis followed by activation of calcium-dependent signaling pathways [43,44] in hASM cells. In the present study, the TNFα-induced increase in pCREB^S133^ and pATF1^S63^ in hASM cells was observed and its association with a downstream signaling cascade affecting mitochondrial biogenesis was illustrated. There is little information available regarding the upstream regulatory kinases for pCREB^S133^ phosphorylation in response to TNFα exposure.

### 3.4. pCREB^S133^ Transcriptionally Activates PGC1α Expression in hASM Cells

The results of the present study showed that pCREB^S133^ binds to the promotor region of the *PGC1α* gene in hASM cells. These results are consistent with previous studies reporting the presence of a CREB binding site in the upstream promoter region of the *PGC1α* gene [23,24,45]. The CREB binding site is highly conserved in humans and mice and regulates the pCREB^S133^-induced transcription of *PGC1α* in different cell types [23,24,45]. The results of the present study confirmed that pCREB^Ser133^ binds to this specific binding site in hASM cells and that 6 h exposure to TNFα increased the promotor occupancy by pCREB^S133^ in *PGC1α* gene promoter in hASM cells. The transcriptional activation by pCREB^S133^ induced by TNFα was also reflected by an increase in PGC1α protein expression.

### 3.5. PGC1α Activates Transcription of NRFs and TFAM in hASM Cells

To further explore the signaling cascade mediated by the TNFα-induced increase in pCREB^S133^ and PGC1α expression, we focused on the signaling pathway downstream to PGC1α, which acts as a major factor regulating mitochondrial biogenesis [12,16]. Previous studies reported that PGC1α directly binds to the regulatory region of the *NRF1* gene and induces mRNA expression [12,16,32,46]. Consistent with this, we found that TNFα induces an increase in *NRF1* mRNA and NRF1 protein levels in hASM cells. NRF1 is known to bind to the upstream regulatory region of the *TFAM* gene and accelerate mtDNA replication [47,48]. We found that TNFα increased *TFAM* gene expression in hASM cells. We also found that TNFα exposure increased *NRF2* mRNA expression. NRF2, like NRF1, binds to the regulatory region of other mitochondrial genes (TOMM20 and several antioxidant genes) and modulates the structure and function of mitochondria [49,50,51,52].

### 3.6. TNFα Induces an Increase in Metabolic Demand in hASM Cells

Previous studies from our lab reported that 24 h TNFα exposure increases force generation and ATP hydrolysis in porcine ASM [10,44]. The increase in force generation in porcine ASM is due to a TNFα-induced increase in contractile protein expression [10,44]. The TNFα-induced increase in ATP consumption also relates to an increase in cross-bridge cycling rate reflecting reduced internal loading [10,44]. Furthermore, TNFα also increases hASM cell proliferation thereby increasing energetic demand [11]. Thus, TNFα increases metabolic demand, which appears to trigger a homeostatic response to increase ATP production by promoting mitochondrial biogenesis and mitochondrial volume density in hASM cells. In addition, by increasing mitochondrial volume density, the demand for ATP production and oxygen consumption per mitochondrion is reduced, thus limiting reactive oxygen species (ROS)-induced oxidative stress [9,11].

### 3.7. Clinical Significance

In the current COVID-19 pandemic, we have become keenly aware of the negative impact of airway inflammation, which is mediated by several pro-inflammatory cytokines including TNFα. The signaling pathways triggered by pro-inflammatory cytokines activate pathophysiological mechanisms underlying respiratory diseases such as asthma, chronic bronchitis, and chronic obstructive pulmonary disease (COPD). Previous studies from our lab demonstrated that TNFα exposure increases metabolic demand in hASM cells consistent with airway hyper-contractility found in many respiratory diseases. We hypothesize that the metabolic/oxidative stress induced by airway inflammation triggers a homeostatic response to increase ATP production via mitochondrial biogenesis and increased mitochondrial volume density. The current study explored a novel mechanism in hASM cells by which TNFα can mitigate the negative impact of inflammation-induced hyper-contractility, ATP consumption, increased ROS production, and oxidative stress-related cell death.

### 3.8. Experimental Limitations and Future Studies

In this study, hASM cells were treated with 20 ng/mL TNFα for 6 h. The serum concentration of TNFα is ~50–100 pg/mL but varies across the patients. Upon inflammatory stimulation, the serum concentration of TNFα may increase up to 5 ng/mL. However, tissue concentration of cytokines are almost certainly higher than serum concentration, especially in inflamed tissue. For this reason, most in vitro studies have used concentrations of TNFα between 10–100 ng/mL [53]. Previous studies from our lab reported that 20 ng/mL TNFα induced an ER stress response involving autophosphorylation of IRE1α and downstream splicing of the transcription factor X-box binding protein 1 (XBP1) [11,54]. TNFα-induced activation of the ER stress response in hASM cells is apparent by 3–6 h exposure. Additionally, the 20 ng/mL TNFα concentration induced ROS production in hASM cells. Importantly, the present study explored the signaling pathway underlying TNFα-induced mitochondrial biogenesis and increased mitochondrial volume density. We selected the 6 h time point to include effects of TNFα on gene expression as well as protein levels. We confirmed that 6 h exposure to 20 ng/mL TNFα induced mitochondrial biogenesis and increased mitochondrial volume density comparable to the effects previously observed at 24 h exposure. The study was not designed to examine gain/loss of function. The mechanism underlying TNFα-induced pCREB^S133^ phosphorylation and involvement of the specific kinases remains unknown and will be the focus of future studies. Future studies may focus on different kinases and their contribution to TNFα-induced pCREB^S133^ phosphorylation in hASM cells. Using specific activators or inhibitors (for gain and loss of function) for pCREB^S133^ phosphorylation would be valuable to further confirm the effect of pCREB^S133^ phosphorylation on this TNFα-induced mitochondrial biogenesis.

## 4. Materials and Methods

### 4.1. Dissociation of Cells from Bronchiolar Tissue

During lung resection surgeries, bronchiolar tissue samples were obtained from anonymous female and male patients ranging in age from 49 to 71 years old who were currently non-smokers with no history of respiratory disease (Figure 9). The research protocol was reviewed by the Institutional Review Board at Mayo Clinic and considered to be exempt (IRB #16-009655). Patient consent was obtained during pre-surgical evaluation. After pathological assessment of the lung tissue, normal regions were identified, and third to sixth generation bronchi were dissected (Figure 9). From the bronchiolar tissue, the smooth muscle layer was further dissected for enzymatic digestion using papain and collagenase and an ovomucoid/albumin separation method was employed for the dissociation of cells following the manufacturer’s instructions (Worthington Biochemical, Lakewood, NJ, USA) [11,54]. Cells were cultured in phenol red-free DMEM/F-12 (Invitrogen, Carlsbad, CA, USA) medium with a pH of 7.4, supplemented with 10% fetal bovine serum (Cat. No. A3840002, Gibco, Thermo Fisher Scientific, Rockford, IL, USA) and maintained at 37 °C, 5% CO_2_, 95% air.

### 4.2. Phenotyping Dissociated hASM Cells

The dissociated bronchiolar cells were phenotyped in two ways depending on experimental use. In cell imaging studies, hASM phenotype was determined by the expression of α-smooth muscle actin (α-SMA) based on immunohistochemistry as previously described [11]. Briefly, dissociated cells were plated (~10,000 cells per well) in Nunc™ Lab-Tek™ II 8-well multi-chamber plastic microscope slides (Cat. No. 154534, Nunc™ Lab-Tek™ II Chamber Slide™, Thermo Fisher Scientific, Rockford, IL, USA) in DMEM/F-12 media. The cells were fixed in 4% paraformaldehyde (Cat. No. 28908, Thermo Fisher Scientific, Rockford, IL, USA) diluted in 1X phosphate buffered saline (PBS) for 10 min at room temperature and blocked for 1 h using a blocking buffer containing 10% normal donkey serum (NDS) (Cat. No. D9663, Sigma Aldrich, St. Louis, MO, USA), 0.2% triton X-100 and 1X PBS at room temperature with gentle agitation to prevent non-specific antibody binding. Cells were incubated overnight at 4 °C with target-specific primary antibodies (α-SMA and fibroblast specific protein 1, FSP1/S100A4) at a dilution of 1:500 in antibody diluent solution (2.5% normal donkey serum, 0.25% sodium azide, 0.2% triton X-100, 1X PBS) (Table 1).

The cells were incubated in species-specific fluorophore-conjugated secondary antibodies (Jackson Immunoresearch, West Grove, PA, USA) at a concentration of 1:400 diluted in antibody diluent at room temperature for 1 h to detect the target proteins. Stained cells were mounted using Fluoro-Gel II mounting medium with 4′,6-Diamidino-2-Phenylindole, Dihydrochloride (DAPI) (Cat. No. 17985-50, Electron Microscopy Sciences, Hatfield, PA, USA) (Thermo Fisher Scientific, Rockford, IL, USA). Dissociated cells were imaged to distinguish α-SMA expressing hASM cells from FSP1 expressing fibroblasts (Figure 2A), and the percentage of hASM cells and fibroblasts were determined as a fraction of total dissociated cells (determined from DAPI). Morphologically, hASM cells were distinctly different from fibroblasts, with a long and fusiform shape compared to smaller fibroblasts (Figure 2A). Based on the single cell-based phenotyping, among the dissociated cell population, hASM cells constituted ~95% of all dissociated cells with the remainder being fibroblasts.

For molecular and biochemical measurements, cell homogenates were used. Cell homogenates were phenotyped by the expression of α-SMA (determined by Western blot) relative to total protein (>2% for inclusion of sample). In brief, protein samples extracted from lung-dissociated cells were denatured and run in SDS-PAGE, followed by transfer to PVDF membrane. The PVDF membrane was probed with α-SMA specific antibody. The α-SMA-specific band intensity was quantified using Image Lab software (version 6.0.1) and normalized to the total protein loaded in the lane (Figure 2B).

### 4.3. Experimental Design

Figure 2 depicts the experimental design used in the present study. From the dissected smooth muscle layer of bronchiolar samples, the dissociated cells were cultured for 1–3 passages. Dissociated cells were only used from culture passages 2–3. Only dissociated bronchiolar cells expressing α-SMA were included in the present study. Based on the absence of α-SMA immunoreactivity and/or α-SMA protein expression (Western blot), samples from three patients were excluded from further analysis. After differentiating cells by serum deprivation for 48 h, the phenotype of hASM cells was confirmed by immunocytochemistry (Figure 2) with >90% displaying the hASM phenotype (i.e., α-SMA expression). In each patient, hASM cells were split into two experimental groups: TNFα-treated and untreated controls. In the TNFα-treated group, hASM cells were exposed to media containing 20 ng/mL TNFα for 6 h. The concentration of TNFα and duration of exposure were selected based on previous studies exploring the dose and time course of TNFα effects on induction of endoplasmic reticulum (ER) stress and mitochondrial remodeling in hASM cells [11,54]. In particular, we found that 24 h TNFα exposure increased mitochondrial biogenesis and mitochondrial volume density in hASM cells. The shorter 6 h exposure period was selected to explore upstream regulatory signaling pathways responsible for the effects on mitochondrial biogenesis and volume density. For example, we previously found that 24 h TNFα (20 ng/mL) exposure increases the cytosolic calcium response to muscarinic stimulation in hASM [10]. We also found that acute (3–6 h) TNFα (20 ng/mL) exposure induced an ER stress response in hASM cells involving phosphorylation of the inositol requiring enzyme 1-alpha (IRE1α) [54]. Based on these previous results, we selected a period of 6 h exposure to 20 ng/mL TNFα.

Mitochondrial DNA (mtDNA) and mRNA expression in muscle homogenates was analyzed by quantitative real-time PCR. Bioinformatic analysis and ChIP was done for gene target identification. Statistical analysis on mitochondrial volume density in individual hASM cells was based on morphometry of six hASM cells per sample times two groups per sample (TNFα-treated or untreated) and five patients (*n* = 30 hASM cells per group) using a one-way ANOVA for repeated measures. Statistical analyses of protein expression, mtDNA copy number, and mRNA expression were based on *n* = 6 hASM samples (patients) using a paired *t*-test. In all analyses, statistical significance was set at * *p* < 0.05.

### 4.4. Protein Extraction and Western Blot

The hASM cells were lysed in 1X Cell Lysis Buffer (Cat. No. 9803, Cell Signaling Technology, Danvers, MA, USA) supplemented with a protease inhibitor cocktail (Cat. No. 11836170001, Roche, Millipore Sigma, Burlington, MA 01803, USA) and phosphatase inhibitors (PhosSTOP, Cat. No. 4906845001, Roche, Millipore Sigma, Burlington, MA 01803, USA). Protein concentrations were quantified using a DC (detergent-compatible) protein assay that utilizes the principle of the well-documented Lowry-based assay (Bio-Rad, Berkeley, CA, USA). An amount of 60–80 mg of total protein from each sample was denatured in 1X Laemmli sample buffer with beta-mercaptoethanol at 100 °C for 3 min. After denaturation, the samples were loaded onto stain-free polyacrylamide gel (Bio-Rad, Berkeley, CA, USA) and run via SDS-PAGE. Total protein content in each lane was visualized, imaged, and analyzed using the ChemiDoc MP Imaging System (Bio-Rad, Berkeley, CA). The proteins from the gel were then transferred to a polyvinylidene difluoride (PVDF) membrane (Bio-Rad, Berkeley, CA, USA) using the Trans-Blot Turbo system (Cat. No. 1704150EDU, Bio-Rad, Berkeley, CA, USA). After transfer, the membranes were blocked using 5% non-fat dry milk to prevent non-specific binding of antibodies followed by overnight incubation with primary antibodies designed to recognize and bind to the protein of interest (Table 1). Horseradish peroxidase conjugated species-specific secondary antibodies were used to detect the primary antibody targets and amplify the signal for easier detection (1:7500 dilution). Bands were developed by incubating the PVDF membrane in chemiluminescent SuperSignal West Dura Extended Duration Substrate (Cat. No. PIA34075, Thermo Fisher Scientific, Rockford, IL) for 3 min and visualized using the ChemiDoc MP Imaging System. Band intensity was quantified using Image Lab software (version 6.0.1) and normalized to the total protein visualized in each lane. For pCREB^S133^ and total CREB detection, the same blot was used to avoid gel-to-gel variation. The blot was probed with rabbit monoclonal pCREB^S133^ specific antibody (Table 1) that also cross-reacts and detects the phosphorylated form of pATF1^S63^. The blots used for pCREB^S133^ and pATF1^S63^ detection were stripped with Restore Western Blot Stripping Buffer (Cat. No. 46428, Thermo Fisher Scientific, Rockford, IL, USA) to remove the pCREB^S133^/pATF1^S63^ specific antibody, and the stripped blot was reprobed with mouse monoclonal CREB specific antibody for total CREB detection (Appendix A). We were unable to validate an antibody for total ATF1. All gel images and full blot images are available in Appendix A).

### 4.5. Labeling and Confocal Imaging of Mitochondria in hASM Cells

For confocal imaging, dissociated cells were plated on m-slide 8-well ibiTreat chambers (Cat. No. 80826, ibidi GmbH, Gräfelfing, Germany). Mitochondria were labeled with 200 nM MitoTracker Green FM (Cat. No. M7514, Thermo Fisher Scientific, Rockford, IL; excitation wavelength: 490 nm; emission wavelength: 516 nm) in serum-free DMEM/F-12 media, pre-warmed at 37 °C for 15 min in the incubator followed by extensive washing with Hanks’ Balanced Salt Solution (HBSS) (Cat. No. H6648, Sigma Aldrich, St. Louis, MO 68178, USA). Mitochondria in hASM cells (distinguished by their size and fusiform shape) were imaged using a Nikon Eclipse A1 laser scanning confocal microscope with a ×60/1.4 NA oil-immersion objective at 12-bit resolution into a 1024×1024-pixel array. The dynamic range for imaging was set by first scanning a region containing no MitoTracker fluorescence signal and then a second region of interest containing maximum MitoTracker fluorescence. A series of 0.5 mm optical slices were acquired for each image. The images obtained were deconvolved using NIS Elements (Nikon Instruments Inc., Melville, NY, USA) to improve signal-to-noise in the images and thereby improve contrast and edge detection. The voxel dimensions of each deconvolved optical slice were 0.207 × 0.207 × 0.5 µm.

Multiple hASM cells were visualized in each microscopic field. Based on an a priori power analysis of variance in mitochondrial volume density measurements in untreated hASM cells, six individual hASM cells per treatment group (TNFα-treated or untreated) from five bronchial samples (patients) were analyzed (*n* = 30).

### 4.6. Measurement of Mitochondrial Volume Density

After deconvolution of each optical slice, the Z-series of images was reconstructed in 3D and the boundaries of each hASM cell were delineated using ImageJ-Fiji software (https://imagej.nih.gov/ij/) (ImageJ 1.53t) (Figure 3A). After background correction and ridge filter detection, mitochondria within hASM cells were identified by thresholding to create a binary image and then skeletonized using the ImageJ mitochondrial analyzer plugin [11,36]. Mitochondrial volume density was calculated as the ratio of mitochondrial volume (number of voxels containing thresholded MitoTracker fluorescence) within the cell to the total volume of the delineated hASM cell [11,34,35,55,56].

### 4.7. Genomic DNA Extraction and Quantification of mtDNA

Genomic DNA was extracted from TNFα-treated hASM cells and untreated control hASM cells using QIAamp DNA Mini Kit (Cat. No. 51304, Qiagen, Hilden, Germany) as per manufacturer’s protocol and quantified using Nanodrop spectrophotometer (Thermo Fisher Scientific, Rockville, IL, USA). The relative copy number of human mtDNA was quantified using Human Mitochondrial DNA Monitoring Primer Set (Cat. no. 7246, Takara Bio USA, Mountain View, CA, USA). Briefly, genomic DNA from TNFα-treated hASM cells and untreated control hASM cells was subjected to quantitative real-time PCR (qPCR) using SYBR green master mix (Cat. No. 04707516001, LightCycler^®^ 480 SYBR Green I Master, Roche Scientific, Thermo Fisher Scientific, Rockville, IL) as per the manufacturer’s instructions. The primer set contains two nuclear gene-specific primers, solute carrier organic anion transporter family member 2B1 (*SLCO2B1*) and serpin family A member 1 (*SERPINA1*)]; and two mtDNA specific primers, NADH dehydrogenase subunit 1 (*ND1*) and NADH: ubiquinone oxidoreductase core subunit 5 (*ND5*)] (Cat. no. 7246, Takara Bio USA, Mountain View, CA, USA). The relative quantification of mtDNA copy number was represented as the difference in cycle threshold (Ct) values for mtDNA and nuclear DNA.

### 4.8. Bioinformatic Analysis for Transcription Factor Binding Site Prediction

Total promoter sequence of PGC1α was obtained from the University of California Santa Cruz (UCSC) Genome Browser, selecting the most updated genome assembly (Human GRCh38/hg38). Transcription factor binding sites for pCREB^S133^ and pATF1^S63^ were predicted using two databases. TF-bind was used as a tool for searching transcription factor binding motifs in target DNA sequence (Figure 7A) that utilizes the weight matrix available in eukaryotic transcription factor database TRANSFAC R.3.4, developed by Dr. Wingender et al. (tfbind.hgc.jp). The transcription factor binding site was further confirmed using Eukaryotic Promoter Database (EPD), which is an annotated non-redundant collection of eukaryotic POL II promoters with experimentally validated transcriptional start sites [57].

### 4.9. Chromatin Immunoprecipitation Assay (ChIP)

Chromatin Immunoprecipitation Assay (ChIP) assay was performed in hASM cells treated with TNFα for 6 h and compared with the untreated control as per the manufacturer’s instruction (Cat. No. 26157, Thermo Fisher Scientific, Rockville, IL). Paraformaldehyde-fixed hASM cells were treated with micrococcal nuclease (MNase) to remove extranuclear DNA, followed by sonication. Sheared chromatin was immunoprecipitated with CREB specific antibody (Cat. No. 4820, Cell Signaling Technology, Danvers, MA, USA) and the antibody–chromatin–protein complex was purified using Protein-A/G beads. We were unable to validate an antibody for total ATF1. In the assays, positive (RNA Polymerase II that targets GAPDH) and negative controls (IgG) were included as per the manufacturer’s instruction. Immunoprecipitated chromatin was eluted and purified, and qPCR was performed to estimate the abundance of CREB at the promoter region of the target gene PGC1α using specific primers (Table 2). Ct values were normalized to the negative control (IgG). Data were interpreted as the ratio of precipitated DNA to the total input of genomic DNA and compared between control and TNFα-treated groups.

### 4.10. RNA Extraction, cDNA Preparation and qPCR

Total RNA was extracted from hASM cells treated with TNFα for 6 h and untreated control using RNeasy extraction kit (Cat. No. 74104, Qiagen, Hilden, Germany) as per the manufacturer’s instruction. In brief, hASM cells were lysed and subjected to ethanol-mediated extraction of RNA. Genomic DNA was removed by an on-column DNase treatment. Extracted RNA samples were quantified using a nanodrop spectrophotometer (Thermo Fisher Scientific, Rockville, IL, USA). A total of 500 ng of RNA was used for complementary DNA (cDNA) synthesis followed by qPCR using specific primer sets (Table 2) to estimate the mRNA expression of target genes (*NRF1*, *NRF2* and *TFAM*).

### 4.11. Statistical Analysis

For each of the experiments, hASM cells were dissociated from bronchial tissue samples obtained from both female and male patients. Sex is an important biological variable, but the study was not powered to detect sex differences. Patient samples were included only if dissociated cells displayed a hASM phenotype as confirmed by the expression of α-SMA by immunocytochemistry and Western blot. hASM cells dissociated from the same patient and same passage (1–3 passages) were split into two groups, TNFα groups and untreated controls. Thus, hASM cells from each patient served as their own controls in assessing the impact of TNFα treatment. The number of samples required to detect a difference of >20% between TNFα-treated vs. untreated control hASM cells were determined by an a priori power analysis (α = 0.05, β = 0.80). Shapiro–Wilk test was employed to confirm a normal distribution in the data. The effect of TNFα treatment on mitochondrial volume density in individual hASM cells was based on measurements from six hASM cells per treatment group (TNFα-treated or untreated) from five bronchial samples (patients) using a one-way ANOVA for repeated measures (*n* = 30). Since hASM cells dissociated from each patient served as their own control, a paired *t*-test was performed for statistical analyses using GraphPad Prism 9 software. Statistical significance was indicated as * *p* < 0.05. All experimental findings were presented in two ways. For comparisons between TNFα-treated and untreated sets within each patient, “symbol and line” scatter plots were used to show individual data points for each group. Each color represents one bronchial sample (patient), squares represent TNFα-treated samples, and circles represent untreated controls. For comparisons between TNFα-treated and untreated controls across the six patients, data were presented as box-and-whisker plots, showing the median and minimum to maximum distribution of the datasets.

## 5. Summary and Conclusions

The results of the present study provide new insight into the mechanisms underlying TNFα-mediated mitochondrial biogenesis in hASM cells. Exposing hASM cells to TNFα for 6 h activates the pCREB^S133^ signaling pathway (Figure 1), involving transcriptional activation of *PGC1*α gene expression. Downstream, PGC1α mediates transcriptional activation of *NRF1*, *NRF2*, and *TFAM* expression, which in turn are responsible for mtDNA synthesis, mitochondrial biogenesis, and an increase in mitochondrial volume density in hASM cells. In previous studies, we found that TNFα induces hypercontractility and an increase in ATP consumption in hASM cells, which is associated with an increase in oxygen consumption rate and ROS production. We conclude that TNFα triggers mitochondrial biogenesis as a homeostatic response to meet the increased ATP demand and offset the oxidative stress related to ROS production.

## Figures and Tables

**Figure 1 ijms-24-05788-f001:**
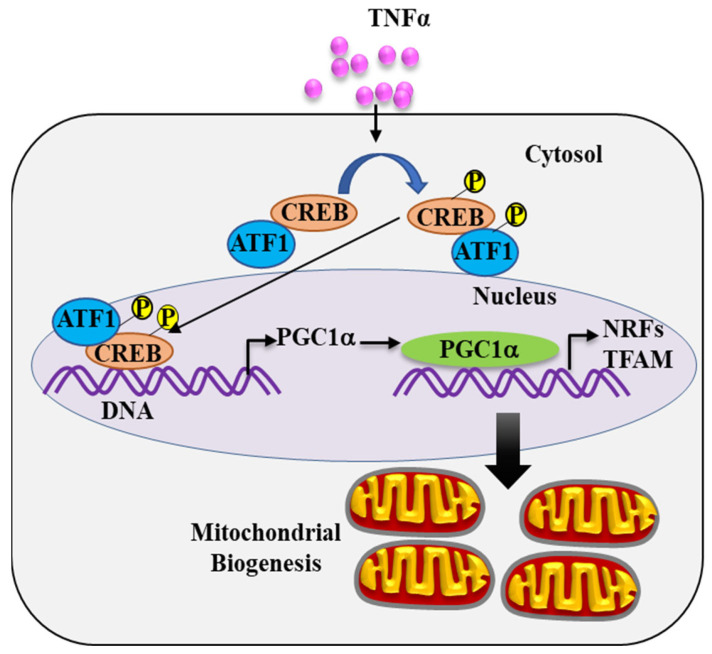
Conceptual framework. We hypothesize that TNFα triggers a signaling cascade involving phosphorylation of the transcription factors cAMP response element binding protein (pCREB^S133^) together with activating transcription factor 1 (pATF1^S63^). pCREB^S133^/pATF1^S63^ binds to the peroxisome proliferator activated receptor gamma co-activator 1 alpha (PGC1α) promoter, thereby increasing PGC1α expression, which acts as a transcriptional co-activator increasing expression of nuclear respiratory factors (NRF1 and NRF2) and transcription factor A, mitochondrial (TFAM), thereby promoting mitochondrial DNA (mtDNA) replication and mitochondrial biogenesis.

**Figure 2 ijms-24-05788-f002:**
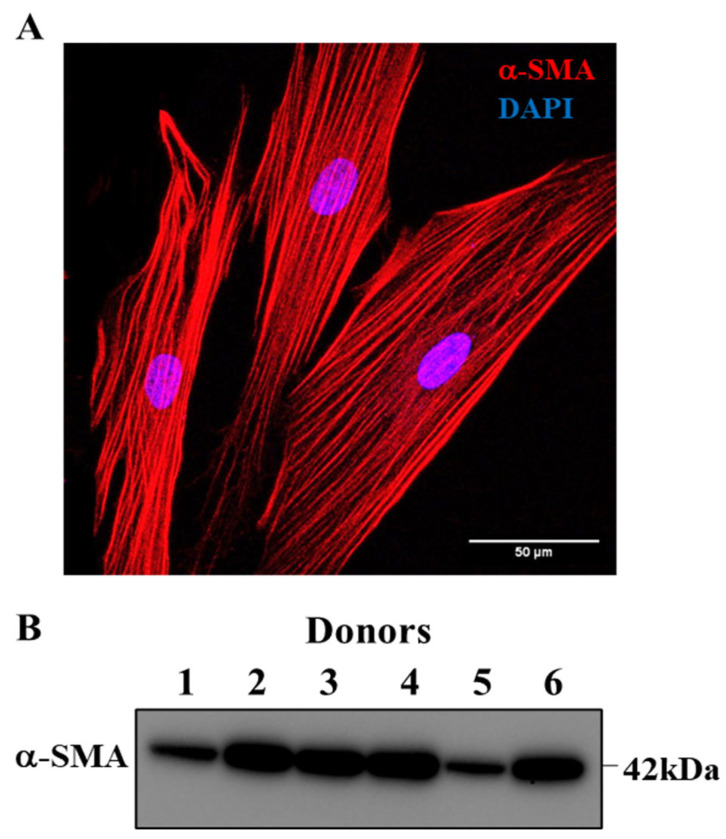
Confirmation of hASM phenotype. (**A**). Representative maximum intensity Z projection image of hASM cells displaying immunoreactivity for alpha-smooth muscle actin (α-SMA) expression, which was used to confirm hASM phenotype. Note that α-SMA immunoreactive hASM cells were larger and displayed a fusiform shape (scale bar = 50 μm). (**B**). Western blot confirming the expression of α-SMA in bronchiolar sample homogenates. Only those bronchiolar samples displaying α-SMA expression were used for biochemical/molecular biology analyses.

**Figure 3 ijms-24-05788-f003:**
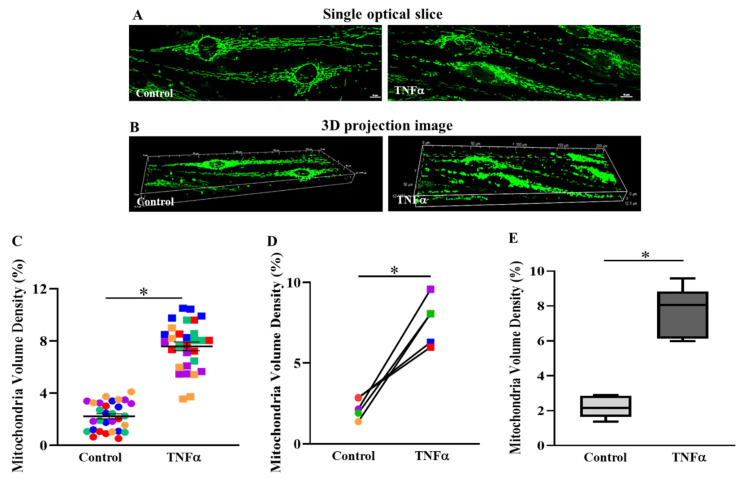
TNFα increases mitochondrial volume density in hASM cells. Mitochondria in hASM cells were labeled with MitoTracker Green and imaged in 3D using confocal microscopy (**A**). Representative confocal optical slice (0.5 mm) image of labeled mitochondria in hASM cells. (**B**). Three-dimensional projection image of a Z-series of optical slices from which mitochondrial morphology and volume density were determined in hASM cells (scale bar  =  10 μm), (**C**). Scatter plot showing mitochondrial volume densities of hASM cells from five patient samples (*n* = 30 hASM cells per group). Note that mitochondrial volume densities were significantly higher in TNFα-treated (20 ng/mL for 6 h) hASM cells compared to untreated control cells. Data are presented as mean ± SEM. Each color represents results from one patient’s bronchial sample, with squares representing TNFα-treated hASM cells and circles representing untreated controls. The effect of TNFα treatment on mitochondrial volume density in individual hASM cells was based on statistical analysis (one-way ANOVA for repeated measures) of mitochondrial volume densities of six hASM cells per treatment group (TNFα-treated or untreated) from five bronchial samples (patients) using a (* *p* < 0.05; *n* = 30). (**D**). Each symbol represents the mean mitochondrial volume density of six hASM cells from each patient with lines showing the change in mitochondrial volume density induced by TNFα treatment. Each color represents results from one patient, with squares representing TNFα-treated hASM cells and circles representing untreated controls (* *p* < 0.05). (**E**). Box-whisker plot showing the median and minimum to maximum distribution of mitochondrial volume densities summarizing results from all five patients (* *p* < 0.05).

**Figure 4 ijms-24-05788-f004:**
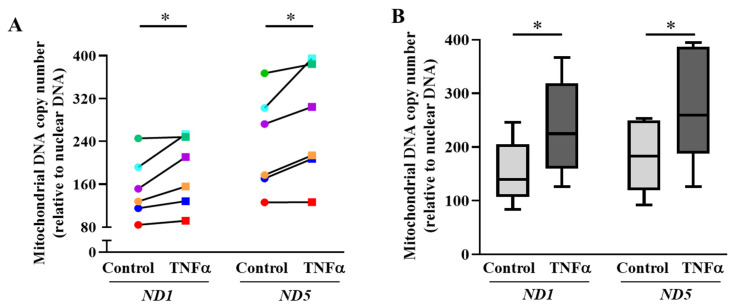
TNFα increases mitochondrial DNA copy number in hASM cells. (**A**). Mitochondrial DNA copy number was quantified using mitochondrial DNA specific primer sets (both *ND1* and *ND5*) via qPCR and represented as a fold change relative to nuclear DNA (*SLCO2B1* and *SERPINA1*). Both mtDNA copy numbers (*ND1* and *ND5*) were compared between TNFα-treated (20 ng/mL for 6 h) (squares) and untreated (circles) hASM cells for each of six patient hASM samples (represented by different colors). Note that in each patient, relative DNA copy numbers (*ND1* and *ND5*) increased in hASM cells exposed to TNFα (20 ng/mL for 6 h) (* *p* < 0.05; *n* = 6). (**B**). The mtDNA copy number results were summarized as box-whisker plots showing the median and minimum to maximum distribution of *ND1* and *ND5* relative mtDNA copy number across the six patient samples. Statistical analyses on measures of mtDNA copy number were performed on *n* = 6 patient hASM samples using a paired *t*-test (* *p* < 0.05).

**Figure 5 ijms-24-05788-f005:**
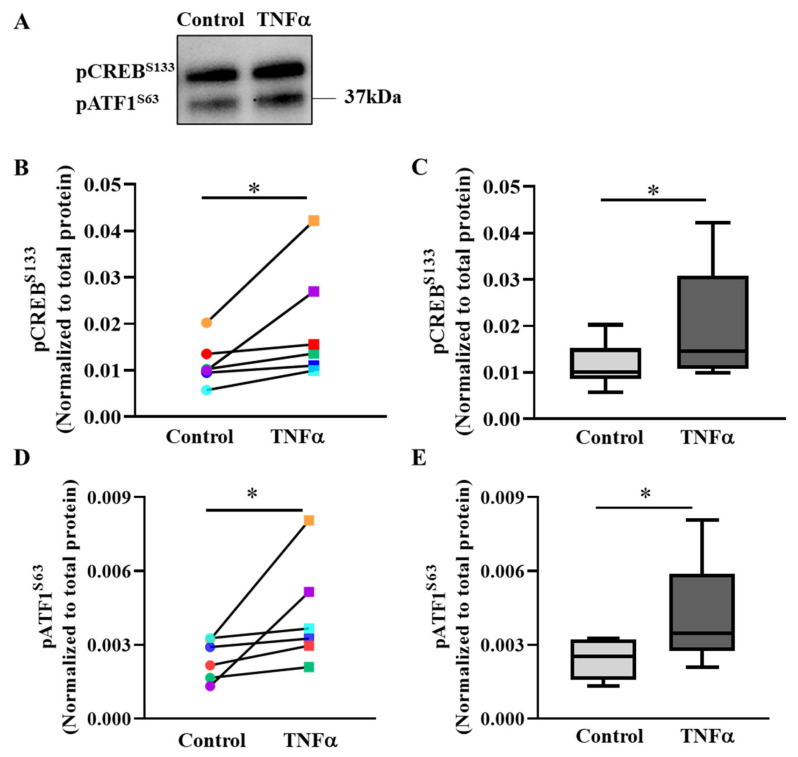
TNFα increases pCREB^S133^ and pATF1^S63^ phosphorylation in hASM cells. (**A**). Representative Western blot showing the effect of exposing hASM cells to TNFα (20 ng/mL for 6 h) as compared to untreated control hASM cells on the expression of pCREB^S133^ and pATF1^S63^. Note that the antibody used to detect pCREB^S133^ also detects pATF1^S63^. The upper band was pCREB^S133^ and the lower band was phosphorylated pATF1^S63^. Expression of pCREB^S133^ and pATF1^S63^ was normalized to total protein loaded in the lane. (**B**). In each of six patient hASM samples (represented by different colors), the relative expression of pCREB^S133^ increased in TNFα-treated (squares) compared to untreated (circles) hASM cells (* *p* < 0.05; *n* = 6). (**C**). The pCREB^S133^ results were summarized as box-whisker plots showing the median and minimum to maximum distribution of normalized pCREB^S133^ expression across the six patient samples (* *p* < 0.05; *n* = 6). (**D**). In each of six patient hASM samples (represented by different colors), the relative expression of pATF1^S63^ increased in TNFα-treated (squares) compared to untreated (circles) hASM cells (* *p* < 0.05; *n* = 6). (**E**). The pATF1^S63^ results were summarized as box-whisker plots showing the median and minimum to maximum distribution of normalized pATF1^S63^ expression across the six patient samples. Statistical analyses on measures of pCREB^S133^ and pATF1^S63^ were performed on *n* = 6 bronchial samples (patients) using a paired *t*-test (* *p* < 0.05; *n* = 6).

**Figure 6 ijms-24-05788-f006:**
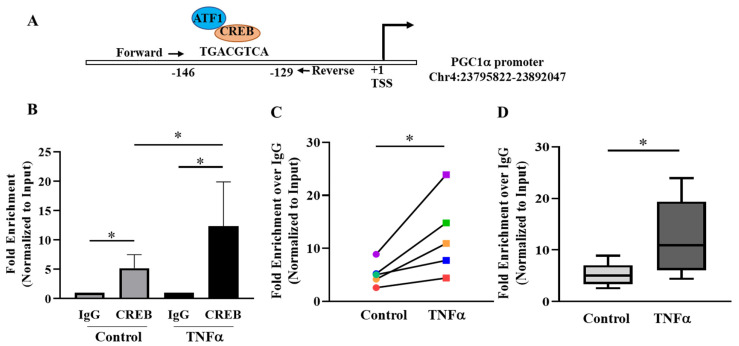
Assessment of PGC1α promoter occupancy by pCREB^S133^ in hASM cells. (**A**) Putative binding site for pCREB^S133^ and pATF1^S63^ in the promoter region of *PGC1*α was predicted by bioinformatics analysis. (**B**) pCREB^Ser133^ binding to the *PGC1*α promoter region was confirmed in hASM cells treated with TNFα (20 ng/mL for 6 h) and untreated hASM cells by chromatin immunoprecipitation assay (ChIP). The data obtained by ChIP-qPCR were represented as the fold enrichment over IgG control and normalized as % of input DNA. (**C**) In each of the five patient hASM samples (represented by different colors), pCREB^Ser133^ binding to the *PGC1*α promoter region was significantly higher in TNFα-treated (squares) compared to untreated (circles) control hASM cells (* *p* < 0.05; *n* = 5). (**D**) The results of pCREB^Ser133^ binding to the *PGC1*α promoter region across the five patient hASM samples were summarized by a box-whisker plot showing the median and minimum to maximum distribution. Statistical analysis of pCREB^Ser133^ binding to the *PGC1*α promoter region was performed on *n* = 5 hASM samples (patients) using a paired *t*-test (* *p* < 0.05).

**Figure 7 ijms-24-05788-f007:**
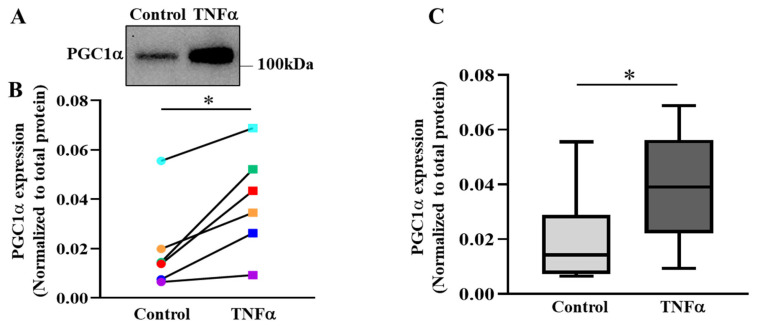
TNFα increases PGC1α protein expression in hASM cells. (**A**) Representative Western blot showing the effect of exposing hASM cells to TNFα (20 ng/mL for 6 h) as compared to untreated control hASM cells on the expression of PGC1α expression hASM cells. (**B**) In each of six patient hASM samples (represented by different colors) TNFα treatment (squares) significantly increased PGC1α expression compared to untreated hASM cells (circles) (* *p* < 0.05; *n* = 6). (**C**) The PGC1α results were summarized as box-whisker plots showing the median and minimum to maximum distribution of normalized PGC1α expression across the six patient samples. Statistical analyses on measures of PGC1α were performed on *n* = 6 bronchial samples (patients) using a paired *t*-test (* *p* < 0.05).

**Figure 8 ijms-24-05788-f008:**
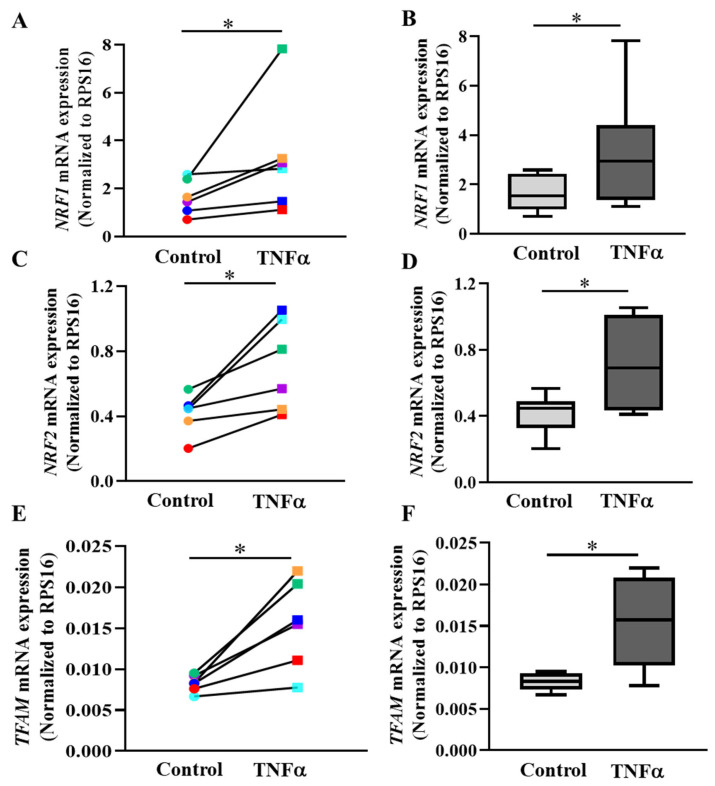
TNFα increases mRNA expression of NRFs and *TFAM* genes in hASM cells. The mRNA expressions of *NRF1* (**A**,**B**)*, NRF2* (**C**,**D**), and *TFAM* (**E**,**F**) were quantified using specific primer sets via qPCR and represented as a fold change relative to RPS16. (**A**,**C**,**E**). The mRNA expression levels of all three genes were compared between TNFα-treated (20 ng/mL for 6 h) (squares) and untreated (circles) hASM cells for each of six patient hASM samples (represented by different colors) (* *p* < 0.05; *n* = 6). (**B**,**D**,**F**). The mRNA expression of *NRF1*, *NRF2* and *TFAM* genes were summarized as box-whisker plots showing the median and minimum to maximum distribution across the six patient samples. Statistical analyses on measures of mtDNA copy number were performed on *n* = 6 patient hASM samples using a paired *t*-test (* *p* < 0.05).

**Figure 9 ijms-24-05788-f009:**
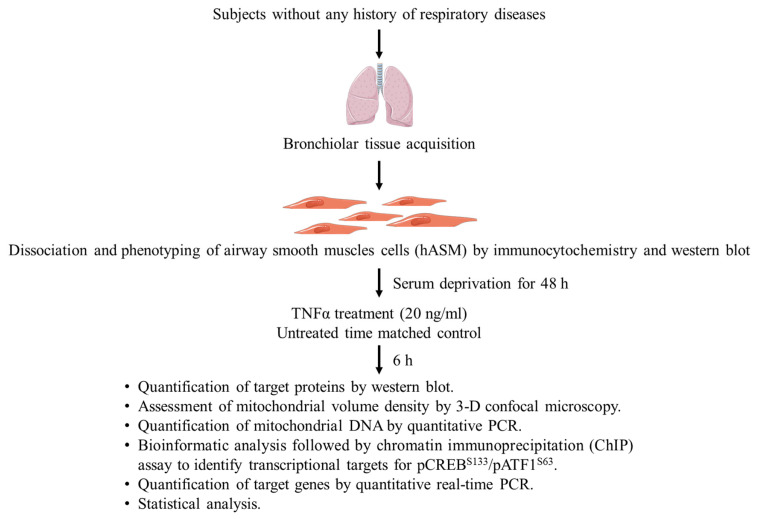
Experimental design. Cells were dissociated from bronchial tissue samples collected during surgery from anonymized female and male patients without any history of respiratory diseases. The hASM phenotype of dissociated cells was confirmed by the expression of alpha-smooth muscle actin (α-SMA) via immunocytochemistry (ICC; cellular analysis) and Western blot (homogenate protein analysis). Dissociated hASM cells from each of six patients undergoing lung surgery (without history of airway diseases or recent smoking) were divided into two experimental groups: (1) treated with TNFα (20 ng/mL for 6 h), or (2) untreated time-matched controls. Mitochondria in hASM cells were labeled using MitoTracker Green and imaged in 3D by confocal microscopy. Protein expression in homogenate samples were analyzed by Western blot.

**Table 1 ijms-24-05788-t001:** List of Primary antibodies used for Western blot and Immunocytochemistry.

Primary Ab	Manufacturer	Catalog Number	Application	Dilution
α-SMA	Abcam, Boston, MA, USA	ab5694	ICC	1:500
FSP1/S100a4	Abcam, Boston, MA, USA	ab124805	ICC	1:500
α-SMA	Abcam, Boston, MA, USA	ab5694	Western blot	1:1000
PGC1α	Novus Biologicals, Centennial, CO, USA	NBP1-04676	Western blot	1:1000
pCREB^S133^/pATF1^S63^	Cell Signaling Technology, Danvers, MA, USA	9198S	Western blot	1:1000
Total CREB	Cell Signaling Technology, Danvers, MA, USA	9104S	Western blot	1:1000
Total CREB	Cell Signaling Technology, Danvers, MA, USA	4820S	ChIP	1:1000
NRF1	Invitrogen, Carlsbad, CA, USA	MA5-32782	Western blot	1:1000

**Table 2 ijms-24-05788-t002:** List of Primers used for qPCR.

Gene Name	Primer Name	Primer Sequence
*NRF1*	NRF1-F	5′-GCAACAGTAGCCACATTGGCT-3′
	NRF1-R	5′-GTCGTCTGGATGGTCATCTCAC-3′
*NRF2*	NRF2-F	5′-CACATCCAGTCAGAAACCAGTGG3′
	NRF2-R	5′-GGAATGTCTGCGCCAAAAGCTG-3′
*PGC1*α	PGC1α -ChIP-F	5′-TGCTTGAAGCCTCCAAAAGT-3′
	PGC1α -ChIP-R	5′-AGTAGGCTGGGCTGTCACTC-3′
*RPS16*	RPS16-F	5′-GTCTGTGCAGGTCTTCGGACGC-3′
	RPS16-R	5′-GACCATTGCCGCGTTTGCAGTG-3′
*TFAM*	TFAM-F	5′-GTGGTTTTCATCTGTCTTGGCAAG-3′
	TFAM-R	5′-TTCCCTCCAACGCTGGGCAATT-3′

## Data Availability

All data are presented in the main manuscript and Appendix A.

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
