# Peer review of "Molecular Mechanisms Underlying TNFα-Induced Mitochondrial Biogenesis in Human Airway Smooth Muscle"

_ijms, 2023, doi:10.3390/ijms24065788_

Round 1
Reviewer 1 Report
General Comments:
This study characterized the effect of TNFα treatment on mitochondrial biogenesis within human airway smooth muscle cells. The authors compared non-treated and 6 hours TNFα treated hASM cells. The authors found that TNFα treatment induces an active phosphorylation of CREB which is responsible for transcriptional expression of PGC1-α and its downstream target genes expression leading to an increased mitochondrial content.
Overall, this study reports interesting novel results about mitochondrial biogenesis in hASM. Increased mitochondrial content following TNFα treatment was previously described but the authors give us a complementary information and demonstration about the mechanistic pathways between inflammation associated to TNFα production and mitochondrial biogenesis. Still, there are a few concerns which need to be addressed.
1- The authors described an increased mitochondrial content using robust and complementary techniques but the mass is not always associated with the functionality. It would be very interesting to analyze whether this increased in mitochondrial mass is associated to a switch in energetic metabolism associated to an increased mitochondrial oxidative metabolism in hASM following inflammation. The authors should perform experiments for phenotyping the mitochondrial functionality and metabolism such as respiration, ATP content or respiratory chain complexes activities.
2- Could the authors analyze ROS levels? Since the authors observed an increased mitochondrial biogenesis, is it associated to an increased mtROS production?
3- Since the authors says that their p-CREB antibody is not specific and cross-reacts with p-ATF1, could the authors redo those experiments using specific antibodies for either p-CREB and p-ATF1? The TNFα treatment seems to affects both phosphorylation of CREB and ATF1, it will be appreciated to clarify those effects.
4- Could the authors analyze the expression of downstream targets genes of PGC1-α at protein level? Since the authors only performed qPCR for NRF1 and NRF2, it will be very interesting and complementary to confirm those results at the protein level.
Author Response
General Comments:
This study characterized the effect of TNFα treatment on mitochondrial biogenesis within human airway smooth muscle cells. The authors compared non-treated and 6 hours TNFα treated hASM cells. The authors found that TNFα treatment induces an active phosphorylation of CREB which is responsible for transcriptional expression of PGC1-α and its downstream target genes expression leading to an increased mitochondrial content.
Overall, this study reports interesting novel results about mitochondrial biogenesis in hASM. Increased mitochondrial content following TNFα treatment was previously described but the authors give us a complementary information and demonstration about the mechanistic pathways between inflammation associated to TNFα production and mitochondrial biogenesis.
Response: We thank the reviewer for the very positive summary and comments.
Still, there are a few concerns which need to be addressed.
Comment 1 - The authors described an increased mitochondrial content using robust and complementary techniques but the mass is not always associated with the functionality. It would be very interesting to analyze whether this increased in mitochondrial mass is associated to a switch in energetic metabolism associated to an increased mitochondrial oxidative metabolism in hASM following inflammation. The authors should perform experiments for phenotyping the mitochondrial functionality and metabolism such as respiration, ATP content or respiratory chain complexes activities.
Response: We agree with the reviewer that an increase in mitochondrial volume density is not always associated with an increase in mitochondrial function. In a previous study, we showed that the 24-h exposure to 20 ng/ml TNFα induced an increase in mitochondrial volume density in hASM cells similar to that induced by 6 h exposure. In this previous study, we reported that the increase in mitochondrial volume density was associated with an overall increase in maximum O2 consumption rate (as measured by Seahorse respirometry). However, when maximum O2 consumption was normalized for the increase in mitochondrial volume density, the maximum O2 consumption rate per mitochondrion in hASM cells was significantly reduced after TNFα exposure compared to untreated cells. In another previous study, we found ATP hydrolysis and force generated by ASM increases following TNFα exposure. Together, these results suggest an adaptive response whereby mitochondrial volume density increases to increase overall cellular O2 consumption and ATP production in order to meet the excess energy demand of contraction (tension cost increase). These points are emphasized in the revised Discussion section.
Comment 2 - Could the authors analyze ROS levels? Since the authors observed an increased mitochondrial biogenesis, is it associated to an increased mtROS production?
Response: We agree with the reviewer that an increase in mitochondrial biogenesis and mitochondrial volume density, as well as an increase in maximum O2 consumption rate (see response above) could be associated with an increase in overall mtROS production. In a previous study, we showed that 24 h TNFα exposure increases mtROS production in hASM cells (measured using MitoSox fluorescence) compared to untreated control. However, when the TNFα-induced increase in overall mtROS production was normalized for the increase in mitochondrial volume density, mtROS production per mitochondrion was reduced. We revised the Discussion section to incorporate these points.
Comment 3 - Since the authors says that their p-CREB antibody is not specific and cross-reacts with p-ATF1, could the authors redo those experiments using specific antibodies for either p-CREB and p-ATF1? The TNFα treatment seems to affects both phosphorylation of CREB and ATF1, it will be appreciated to clarify those effects.
Response: We agree with the reviewer that the antibody used in our study detects both pCREBS133 and pATF1S63 due to the sequence homology and presence of a homologous phosphorylation site. However, we were unable to validate a commercially available antibody that recognizes only pCREBS133 or pATF1S63. While the present study focused on pCREBS133, we believe that the simultaneous effect of TNFα exposure on the phosphorylation of pCREBS133 and pATF1S63 is intriguing since they are reported to act as co-activators of gene transcription. We revised the manuscript to recognize that TNFa induces phosphorylation of both pCREBS133 and pATF1S63 and now discuss their roles as co-activators of gene transcription.
Comment 4 - Could the authors analyze the expression of downstream targets genes of PGC1-α at protein level? Since the authors only performed qPCR for NRF1 and NRF2, it will be very interesting and complementary to confirm those results at the protein level.
Response: We performed an analysis of NRF1 protein levels since we were able to validate an NRF1 antibody. NRF1 protein levels increased in hASM cells after TNFa exposure (Supplementary figure 5). However, we were unable to validate an antibody for NRF2. We revised the manuscript to include these points.
Reviewer 2 Report
This study extended previous work by the authors that reported increased mitochondrial biogenesis in human ASM cells in response to TNF. The current study focused on understanding the mechanisms of TNF induced mitochondrial biogenesis and has provided evidence that this increase is associated with elevated pCREBS133 and PCG1a expression.
Overall, this is an elegantly designed study and experiments are well performed and appropriately analysed, including high quality imaging of stained mitochondria. The authors thoroughly investigated gene expression and transcriptional activation of target genes in primary hASM cells in response to TNF using good methodology. My only slight concern is the small number of patients used in this work (n=6), especially as there is considerable variability in the magnitude of the individual patient response to TNF (eg. Figures 5A&B, 6B&D, 9A). Furthermore, the baseline of unstimulated hASM cells is also quite variable between patients (eg. Fig 5A, 8B).
Author Response
This study extended previous work by the authors that reported increased mitochondrial biogenesis in human ASM cells in response to TNF. The current study focused on understanding the mechanisms of TNF induced mitochondrial biogenesis and has provided evidence that this
increase is associated with elevated pCREBS133 and PCG1a expression.
Overall, this is an elegantly designed study and experiments are well
performed and appropriately analysed, including high quality imaging of
stained mitochondria. The authors thoroughly investigated gene
expression and transcriptional activation of target genes in primary
hASM cells in response to TNF using good methodology.
Response: We thank the reviewer for the summary of our results and the positive comments.
Comment 1. My only slight concern is the small number of patients used in this work (n=6),
especially as there is considerable variability in the magnitude of the
individual patient response to TNF (eg. Figures 5A&B, 6B&D, 9A).
Furthermore, the baseline of unstimulated hASM cells is also quite
variable between patients (eg. Fig 5A, 8B)."
Response: In our experimental/ statistical design, differences across patient samples were considered a random effect variable. Importantly, hASM cells from the same patient were split into two groups: 1) TNFa, and 2) untreated, and an ANOVA (essentially a paired t-test) was used to parse out the source of variability. The number of patients was determined by a power analysis of the primary outcome measures with a b=0.80 and an a=0.05 for indicating a significant difference after TNFa exposure within a patient. In addition to using hASM cells from the same patient, we also used cells from the same passage to reduce any effect of multiple passaging of cells. We revised the Methods section to clarify the experimental design and statistical analysis.
Reviewer 3 Report
The article “Molecular Mechanisms Underlying TNFa Induced Mitochondrial Biogenesis in Human Airway Smooth Muscle” by Debanjali Dasgupta et al. describes the effects and mechanisms of TNFalpha on mitochondria of airway smooth muscles. The study is elegant, well presented and the results are discussed in a clear way. Moreover, a relationship with current clinical problems is presented. I do not have any significant remarks and suggest to accept the paper in the present form.
Author Response
Comment: The article “Molecular Mechanisms Underlying TNFa Induced Mitochondrial Biogenesis in Human Airway Smooth Muscle” by Debanjali Dasgupta et al. describes the effects and mechanisms of TNFalpha on mitochondria of airway smooth muscles. The study is elegant, well presented and the results are discussed in a clear way. Moreover, a relationship with current clinical problems is presented. I do not have any significant remarks and suggest to accept the paper in the present form.
Response: We thank the reviewer for the very positive comments.
Reviewer 4 Report
TNFalpha driven airway inflammation is a major contributor to the development of respiratory diseases. Following on their previous work demonstrating chorinc exposure to TNFalpha may enhance mitochondrial biogenesis in human airway smooth muscle cells, the authors sought to investigate the underlying mechanism responsible for this observation. In isolated primary human airway smooth muscle cells, the authors found that TNFalpha increased mitochondrial density and DNA copy number as early as 6 hours. This was accompanied by increased phosphorylation of CREB and ATF1, as well as their presence at the promoter region of PGC1alpha. This resulted in PGC1alpha and the activation of the downstream signalling molecules that are known to regulate mitochondrial biogenesis. The authors concluded that TNFalpha increases mitochondrial volume density in human airway smooth muscle cells via a CREB/ PGC1alpha mediated pathway.
Major remarks:
1. The rationale for using the selected dosage of TNFalpha on the human airway smooth muscle cells needs to be stated. Is 20 ng/ml for 6 hours mimicking the environment in COPD, asthma, or other lung disease?
2. Is the TNFalpha-mediated mitochondrial biogenesis in the human airway smooth muscle cells considered adaptive or maladaptive? This should be discussed.
3. Was the data tested for normality and how?
4. For the ChIP, was no antibody control included? If so, please include as supplementary.
5. Did the total protein expression of CREB and ATF1 change by TNFalpha? This should be included into the relevant figure.
6. The limitations of the present study should be included in the discussion, as there is no inhibitor, gain-/loss-of function data on CREB.
Minor comments:
1. Page 2, grammatical error “As shown in Fig 1, we hypothesize that TNFalpha induces pCREBS133 phosphorylation in hASM cells transcriptionally activating PGC1alpha…”.
2. Page 2, full stop after “Patient consent was obtained during pre-surgical evaluation”.
3. Page 6, ChIP, the primer sequence for the CREBP promoter region of the PGC1alpha gene should be included.
4. Page 10, “Figure 6c” should be in capital.
Author Response
TNFalpha driven airway inflammation is a major contributor to the development of respiratory diseases. Following on their previous work demonstrating chorinc exposure to TNFalpha may enhance mitochondrial biogenesis in human airway smooth muscle cells, the authors sought to investigate the underlying mechanism responsible for this observation. In isolated primary human airway smooth muscle cells, the authors found that TNFalpha increased mitochondrial density and DNA copy number as early as 6 hours. This was accompanied by increased phosphorylation of CREB and ATF1, as well as their presence at the promoter region of PGC1alpha. This resulted in PGC1alpha and the activation of the downstream signalling molecules that are known to regulate mitochondrial biogenesis. The authors concluded that TNFalpha increases mitochondrial volume density in human airway smooth muscle cells via a CREB/ PGC1alpha mediated pathway.
Response: We thank the reviewer for the precise summary of our study.
Major remarks:
Comment 1. The rationale for using the selected dosage of TNFalpha on the human airway smooth muscle cells needs to be stated. Is 20 ng/ml for 6 hours mimicking the environment in COPD, asthma, or other lung disease?
Response: We revised our manuscript to include the rationale for using the 20 ng/ml concentration of TNFa. Serum TNFa levels vary across patients, but range around 50-100 pg/ml. In inflammatory diseases, serum TNFa levels are elevated with some reports of concentrations of ~5 ng/ml. However, serum levels do not accurately reflect tissue concentrations of TNFa, which may be orders of magnitude higher. Accordingly, in in vitro studies, TNFa exposure ranging from 10 to 100 ng/ml have been used in ASM preparations. In a previous study, we examined the concentration-dependent effect of TNFa exposure on activation of the IRE1a/XBP1s endoplasmic reticulum (ER) stress pathway using concentrations up to 100 ng/ml. Based on an optimal concentration of 20 ng/ml found in these results, we also assessed the dependency of the ER stress response on exposure times ranging from 1 to 48 h. Based on these previous results, we selected a TNFa concentration of 20 ng/ml and an exposure time of 6 h for the experiments in the present study. We revised the Methods to provide the justification for TNFa concentration and exposure time, and this is also discussed in the Discussion section.
Comment 2. Is the TNFalpha-mediated mitochondrial biogenesis in the human airway smooth muscle cells considered adaptive or maladaptive? This should be discussed.
Response: Our previous findings support the hypothesis that TNFα induced mitochondrial biogenesis and increased mitochondrial volume density is an adaptive response by increasing overall O2 consumption and ATP production to meet the increased ATP consumption related to ASM contraction (increased tension cost) while reducing the impact of increased O2 consumption on ROS formation per mitochondrion. We revised the Discussion to include this important point.
Comment 3. Was the data tested for normality and how?
Response: All data were tested for normality using the Shapiro-Wilk test available in the GraphPad Prism software. We revised our manuscript to clarify this point.
Comment 4. For the ChIP, was no antibody control included? If so, please include as supplementary.
Response: We used rabbit IgG as an antibody control per the manufacturer’s recommendation. For the analysis of ChIP data, we analyzed the fold enrichment over the IgG control to negate the non-specific binding of the antibody. We included this in the Methods section and relevant IgG data are now presented in the revised Figure 7.
Comment 5. Did the total protein expression of CREB and ATF1 change by TNFalpha? This should be included into the relevant figure.
Response: We were able to measure total CREB protein levels using a validated antibody. Total CREB was not significantly affected by TNFa exposure. We revised our manuscript and now include a supplementary figure for total CREB (Supplementary figure 3). Unfortunately, we were unable to validate an antibody for total ATF1.
Comment 6. The limitations of the present study should be included in the discussion, as there is no inhibitor, gain-/loss-of function data on CREB.
Response: We agree with the reviewer and have revised the Discussion accordingly to include a section on experimental limitations.
Minor comments:
Comment 1. Page 2, grammatical error “As shown in Fig 1, we hypothesize that TNFalpha induces pCREBS133 phosphorylation in hASM cells transcriptionally activating PGC1alpha…”.
Response: This sentence was rewritten.
Comment 2. Page 2, full stop after “Patient consent was obtained during pre-surgical evaluation”.
Response: We corrected this sentence.
Comment 3. Page 6, ChIP, the primer sequence for the CREBP promoter region of the PGC1alpha gene should be included.
Response: We added the primer sequences in Table 2.
Comment 4. Page 10, “Figure 6c” should be in capital.
Response: We revised “Figure 6C”.